# Outcomes and Follow-Up after Hepatitis C Eradication with Direct-Acting Antivirals

**DOI:** 10.3390/jcm12062195

**Published:** 2023-03-12

**Authors:** Erica Nicola Lynch, Francesco Paolo Russo

**Affiliations:** 1Department of Surgery, Oncology and Gastroenterology, University of Padua, 35122 Padova, Italy; 2Department of Medical Biotechnologies, University of Siena, 53100 Siena, Italy; 3Gastroenterology Research Unit, Department of Experimental and Clinical Biomedical Sciences “Mario Serio”, University of Florence, 50134 Florence, Italy

**Keywords:** viral hepatitis, hepatitis C, direct-acting antivirals, outcomes, management

## Abstract

Treatment of hepatitis C (HCV) has been revolutionized with the introduction of direct-acting antivirals (DAAs). Patients can be treated at more advanced stages of liver disease, with a growing number of cirrhotic patients achieving sustained virological response (SVR). Long-term outcomes for cured patients and the optimal follow-up care of patients after SVR are yet to be defined, because most studies on cirrhotic patients cured with DAAs have a short follow-up period. There are many open questions related to patient management after viral eradication with DAAs, such as which could be the most reliable non-invasive tool to predict liver-related complications, or to what extent viral eradication reduces the risk of liver disease progression in the long term. Growing evidence supports the personalization of follow-up care based on individual risk. The aim of this narrative review is to analyze the impact of viral eradication with DAAs on clinically significant portal hypertension, hepatocellular carcinoma, and extrahepatic manifestations, as well as to summarize indications for optimal follow-up care of HCV patients treated with DAAs.

## 1. Introduction

Hepatitis C virus (HCV) infection remains a global health problem, with 58 million chronically infected people and 400,000 deaths/year according to the World Health Organization (WHO) estimates in 2019 [1]. Most infected patients develop chronic hepatitis, which can determine a spectrum of clinical manifestations, from only minimal necro-inflammatory lesions up to cirrhosis and its complications [2]. Without treatment, HCV infection leads to cirrhosis in 10–20% of cases over a period of 20–30 years. Moreover, in untreated patients, the annual risk of hepatocarcinoma (HCC) and hepatic decompensation is 1–5% and 3–6%, respectively [3]. Spontaneous elimination of HCV in chronic carriers is considered to be rare (2–3.7%) [4,5], although higher rates (19.4%) have been reported, possibly due to a different genetic background or age distribution [6]. Transfusion-transmitted infection and tumor-related liver failure were also found to be associated with viral clearance [4,5]. For many years, interferon (IFN)-based therapy, the only available treatment for HCV infection, offered unsatisfying results, with an unfavorable tolerability profile.

The clinical scenario of HCV infection has been transformed after the introduction of direct-acting antivirals (DAAs). DAAs are highly effective and have a good safety profile, allowing viral eradication even in patients with decompensated cirrhosis. For this reason, there is a growing number of patients who achieve sustained virological response (SVR—i.e., undetectable HCV RNA 12 or 24 weeks after treatment [2]) even at an advanced stage of liver disease. Long-term outcomes and optimal management of HCV patients treated with DAAs are yet to be defined, especially for those cured of infection when already considered cirrhotic. The aim of this narrative review is to analyze the impact of viral eradication with DAAs on clinically significant portal hypertension (CSPH), hepatocellular carcinoma, and extrahepatic manifestations, and to summarize indications for optimal follow-up care of HCV patients treated with DAAs.

## 2. The DAA Revolution

HCV is an RNA virus and does not integrate into the host genome, thus allowing complete eradication after treatment. The first approved therapy for chronic HCV infection was IFN in 1990 [7,8]. Interferon only led to SVR in <10% of cases [9]. In the following years, the association with ribavirin and the use of pegylated IFN increased SVR rates to about 40–50% for genotypes 1 and 4, and to approximately 80% for genotypes 2 and 3 [10]. In 2011, the first two DAAs were introduced (boceprevir and telaprevir), marking the beginning of revolutionary advances in HCV treatment [11]. Boceprevir and telaprevir had serious limitations that were addressed by the following generations of DAAs: they needed to be administered with pegylated IFN, they had an inconvenient dosing schedule, and they were associated with severe adverse effects, such as anaemia, skin rash, and dysgeusia [11].

With the succeeding generations of DAAs, SVR rates reached >95% for all genotypes [12] in non-cirrhotic patients, and 80–90% in cirrhotic patients [13]. DAAs also allowed the treatment of patients who were excluded from IFN-based regimens, such as patients with decompensated cirrhosis, autoimmune, and psychiatric diseases. Genetic variability of HCV was challenging and led to the development of different treatment combinations [12]. In the end, an in-depth understanding of the HCV life cycle allowed the identification of high-barrier compounds with pan-genotypic efficacy [11]. At present, pan-genotypic fixed-dose DAAs with a treatment duration of 8–12 weeks represent the established standard of care for HCV infection [2]. There are only a few remaining treatment challenges in the DAA era. DAA failure was a weak point for an otherwise outstanding treatment. This issue was largely solved with the approval of sofosbuvir-velpatasvir-voxilaprevir [14]. There is a small but growing population of patients resistant to this treatment option who benefit from longer DAA treatment duration in combination with ribavirin, although recommendations are only based on small case series [2,11]. There has also been an extensive debate on whether patients enlisted for liver transplantation (LT) for decompensated cirrhosis should receive DAA therapy before or after transplantation. Current guidelines of the European Association for the Study of the Liver (EASL) recommend treating patients with a Model for End-Stage Liver Disease (MELD) score up to 18–20 before transplantation and deferring viral eradication in more severe patients [2]. The American Association for the Study of Liver Diseases (AASLD) guidelines state that HCV patients with decompensated cirrhosis should be referred to a specialized medical practitioner, ideally in a liver transplant center [15]. In line with the European guidelines, the authors additionally state that HCV patients with a MELD score > 20 or severe portal hypertension complications may be better served by transplantation than antiviral treatment. Finally, the Asian Pacific Association for the Study of the Liver (APALS) guidelines state that DAAs may best be introduced post-transplant rather than pre-transplant in patients with very high MELD scores, without specifying an exact cut-off score [16].

Although there are still issues that present a matter for discussion among experts, there is no doubt that DAAs have profoundly changed the epidemiology of HCV infection.

In 2016, the WHO adopted a strategy which proposed the elimination of viral hepatitis as a public health threat by 2030 (defined as a 90% reduction in new chronic infections and a 65% reduction in mortality, compared with 2015 rates) [1]. Screening strategies should be implemented and regional disparities concerning access to therapy should be addressed in order to fully profit from these revolutionary HCV therapies and reach the ambitious WHO goal of HCV elimination [1].

## 3. Impact on Clinically Significant Portal Hypertension

Successful treatment of HCV infection with interferon-based regimens reduces the hepatic venous pressure gradient (HVPG) and decreases the long-term risk of gastro-esophageal varices [17,18]. As expected, positive results have also been obtained with IFN-free regimens, although it should be noted that the risk of hepatic decompensation persists even after viral eradication with DAAs [19]. This can be at least partly explained by the fact that DAA-based treatments can be administered to patients who already have portal hypertension as well as to patients with a positive history of hepatic decompensation. In order to assess the risk of first hepatic decompensation after viral eradication with DAA, Mendizabal et al. conducted a prospective study on a large cohort derived from the Latin American Liver Research, Educational and Awareness Network (LALREAN) [20]. They enrolled 1760 patients with no previous history of liver decompensation, HCC, or LT who were treated with DAAs and followed up for a median of 26.2 months. Of all patients who achieved SVR, 3.6% (95% CI, 2.7–4.7%) suffered from disease progression (hepatic decompensation, de novo HCC, LT, or death) during the follow-up period. SVR reduced the risk of disease progression by 80% compared to treatment failure. In particular, achieving viral eradication significantly reduced but did not eliminate the risk of liver decompensation [HR 0.3 (CI, 0.1–0.8) *p* = 0.016]. No sub-analysis was performed to identify the baseline characteristics that favored hepatic decompensation after SVR but advanced fibrosis, CSPH, and albumin levels < 3.5 mg/dL were found to be risk factors for liver disease progression. It is therefore understandable that patients with more advanced disease stages are still at risk of hepatic decompensation even after SVR, due to other decompensating factors of virus-related liver damage. 

In order to find possible elements related to disease progression in virologically cured patients, Montaldo et al. studied extracellular vesicle (EV) modification in a small sample of healthy donors (HD) and HCV-infected patients, before and after treatment (15 HD and 16 HCV patients for functional and miRNA cargo analyses, and 17 HD and 23 HCV patients for proteomic and Western blot analyses) [21]. EV from HD seem to have an antifibrogenic ability which is lost in HCV patients even after obtaining SVR, as suggested by previous studies [22,23]. Further studies are needed to investigate the role of serum EV content as a possible biomarker of liver fibrogenesis. 

Similarly, Pons et al. conducted a study to identify non-invasive predictors of liver disease progression in HCV-eradicated patients with compensated advanced chronic liver disease (cACLD), as defined in Baveno VI (liver stiffness measurement [LSM] ≥ 10 kPa and no prior hepatic decompensation) [24]. Early non-invasive prognostic biomarkers able to assess long-term clinical outcomes in HCV cured patients would definitely be useful tools to guide patient management. In this study, the incidence rate of portal hypertension-related decompensation was 0.34/100 patient-years. All patients who suffered liver decompensation (5/572) had a baseline LSM ≥ 20 kPa.

Recent evidence has shown that CSPH can improve or even regress after viral eradication with DAAs [25,26]. Unfortunately, studies are heterogeneous with regard to the definition of CSPH and only a few studies assessed the modification of CSPH after DAA treatment using the gold-standard measurement of HVPG (CSPH = HVPG ≥ 10 mmHg) [26,27]. In the multicenter prospective study by Lens et al., 53–65% of patients still had HVPG values compatible with CSPH two years after SVR, and liver stiffness did not appear to correlate with HVPG reduction after SVR [26]. Notably, 17% of patients showed an HVPG increase at SVR 24. Baseline HVPG ≥ 16 mmHg and previous history of ascites identified the high-risk group for hepatic decompensation after SVR. Semmler et al., on the contrary, support the use of LSM to identify patients at risk of developing complications related to portal hypertension [28]. In their study, the correlation between LSM and HVPG increased from pre- to post-treatment (r = 0.45 vs. 0.60). Post-treatment LSM < 12 kPa and PLT > 150 G/L excluded CSPH (sensitivity: 99.2%), while LSM ≥ 25 kPa was highly specific for CSPH (93.6%). These results were validated in a cohort of cACLD patients. Spleen stiffness does not decrease after viral eradication, as opposed to liver stiffness [29], and it seems to be able to predict the risk of HCC [30]. Thus, the association between spleen stiffness and HVPG after viral eradication should be investigated to assess the potential role of spleen stiffness in the surveillance of CSPH after SVR. Studies on larger cohorts and longer follow-up periods are warranted to assess the true impact of HCV eradication with DAAs on clinically significant portal hypertension.

## 4. Impact on Hepatocellular Carcinoma

### 4.1. HCC Development

In cirrhotic patients with ongoing HCV infection, the incidence of HCC ranges from 1% to 7% per year [31]. Interferon-based treatment regimens have been demonstrated to significantly reduce the risk of tumor development by 77% [32]. In 2017, a comprehensive systematic review, meta-analysis, and metaregression by Waziry et al. demonstrated a similar impact of IFN- and DAA-based therapies on HCC incidence among patients with cirrhosis [33]. As DAAs allow patients to be cured at more advanced stages of liver disease and with higher cure rates, the impact of DAAs on HCC incidence is expected to be even greater compared to IFN-based therapy. However, head-to-head studies are considered unethical precisely due to the exceptional tolerability and efficacy of DAAs which broaden the population of treated patients. The incidence of HCC in patients with advanced compensated liver disease (ACLD)/cirrhosis cured with DAAs ranges from 1.5 to 3.6/100 patient-years, and the risk increases sixfold in patients with CSPH [34].

In recent years, research activity has been directed at identifying factors associated with HCC development post SVR. Absence of portal hypertension, higher albumin and baseline platelet levels were found to be protective against carcinogenesis [35]. All previously mentioned factors could simply bring together patients at less advanced stages of liver disease, but cut-off values were sought to identify the patient population who needed to undergo long-term surveillance. In a study by Degasperi et al., diabetes was an independent predictor of HCC incidence in 565 DAA-treated cirrhotic patients, which could be explained by the increased risk of liver disease progression in diabetic patients [36]. Pons et al. found that patients with LSM ≥ 20 kPa at follow-up and those with LSM 10–20 kPa and albumin levels < 4.4 g/dL made up the high-risk group for HCC development (HCC incidence rate ≥ 1.9/100 patient-years) who needed surveillance [24]. Due to the retrospective design of the study, the time of follow-up measurements was not standardized. In a recent study by Semmler et al., simple algorithms based on factors identifying the liver disease status (albumin, alpha-fetoprotein, liver stiffness measurement) and risk of progression (alcohol consumption) and age (a strong driver of carcinogenesis) were used to stratify patients according to the risk of liver cancer development [34]. Among patients with advanced liver disease, these algorithms were able to recognize those who were at very low risk (<1%/year) of HCC development (approximately two-thirds) [34]. For these patients, surveillance may not be cost-effective, and the authors advocate that personalized surveillance strategies are warranted. In fact, European guidelines recommend a six-monthly ultrasound surveillance for all patients who were F3 or F4 before HCV eradication [2]. A recent Japanese study highlights the underestimated risk of loss of patient adherence to follow-up visits after SVR [37].

### 4.2. HCC Recurrence

Interferon-based therapies lead to modestly reduced rates of HCC recurrence after curative tumor treatment [38]. This benefit was maintained even when later (>two years) recurrences were considered [38]. In vitro and in vivo studies suggested a possible additional role of interferon in the inhibition of carcinogenesis [39]. Soon after the introduction of DAAs, in 2016, a small-cohort study by Reig et al. reported an unexpectedly high rate of tumor recurrence coinciding with HCV eradication, with an increased number of infiltrative or multinodular cases. This was tentatively explained by the fact that an abrupt resolution of inflammation could inhibit the immune system control on tumor progression [40]. In the same year, other small-scaled retrospective studies were published which supported the possible association of DAAs with increased risk of tumor occurrence/recurrence [41,42]. The authors agreed that a direct oncogenic effect of DAAs was highly improbable and that the immune response attenuation after HCV clearance could be the underlying causative mechanism. In the following years, larger studies showed no evidence of increased tumor aggressiveness in de novo or recurrent HCC cases after DAAs [43], and it was suggested that the higher HCC rate in DAA-treated patients could have been related to the fact that DAAs were used in patients at more advanced stages of liver disease and with an inherently higher cancer risk [44,45,46]. Furthermore, meta-analyses also showed that HCC occurrence/recurrence risk in DAA-treated patients was comparable to that of IFN-based therapies [33,47]. In a recent meta-analysis by Sapena et al. using individual patient data from 977 consecutive patients and 328 propensity score-matched controls from the ITA.LI.CA. cohort, the authors examined the rate of HCC recurrence in DAA-treated patients [48]. This study finally put an end to the controversy, as it revealed that there is no association between DAA treatment and a higher HCC recurrence rate. On the other hand, it also showed that patients treated with DAAs remain at risk of developing HCC, and that DAAs do not necessarily lead to improved survival rates when administered in patients already treated for HCC [48,49].

## 5. Impact on Extrahepatic Manifestations

Patients with chronic HCV infection additionally have an increased mortality risk for extrahepatic causes, such as non-Hodgkin’s lymphoma (NHL), chronic kidney disease, and cardiovascular diseases. In a Taiwanese study which included 1095 anti-HCV seropositive vs. 18,541 seronegative patients, anti-HCV seropositivity led to an increased mortality risk from extrahepatic diseases with a hazard ratio of 1.35 (1.15–1.57) [50]. Achieving SVR with DAAs is associated with a significant decrease in all-cause mortality [51].

### 5.1. Cardiovascular Disease

HCV infection is associated with an increased risk of cardiovascular disease-related death and cerebrovascular events (OR, 1.65; 95% CI, 1.07–2.56; *p* = 0.02; OR, 1.30; 95% CI, 1.10–1.55; *p* = 0.002), especially in patients with other risk factors such as diabetes or hypertension [52]. The plausible causative mechanisms which have been identified are: metabolic alterations related to HCV infection, the profibrogenic and inflammatory status linked to HCV infection, as well as potential direct viral mechanisms [53]. However, studies on the topic are highly heterogeneous and need confirmation with prospective research. Two French studies on large cohorts have shown that SVR reduces the number of cardiovascular events [54,55]. Viral eradication with DAAs has also been found to reduce the risk of major cardiovascular events by 67.6% in a population of patients with a prediabetic condition, regardless of the degree of fibrosis [56]. These results are in line with those obtained from the RESIST-HCV cohort, showing that SVR was associated with a reduced risk of cardiovascular mortality regardless of the presence of cirrhosis (HR 0.07, beta-2.67, *p* < 0.001) [57]. A small-scaled study suggests that there may be a shift to a less atherogenic lipid profile after viral eradication with DAAs, due to an increased antioxidant capacity of HDL and an increase in LDL-C/apoB ratio [58].

### 5.2. Insulin Resistance

HCV infection is associated with diabetes in 5.9–43.2% of patients [59,60,61]. HCV induces insulin resistance through various pathogenic processes: oxidative stress, production of inflammatory cytokines, pancreatic ß-cell dysfunction, direct inhibition of insulin signaling [62]. Historically, diabetes and insulin resistance were considered negative predictors of a response to interferon-based HCV treatment [63,64]. On the contrary, DAAs appear to be as effective in diabetic as in non-diabetic patients [62]. Viral eradication appears to improve the metabolism of glucose, even in patients with cirrhosis [65,66]. Diabetic patients are more at risk of developing HCC after SVR, probably due to concurrent metabolic steatohepatitis or reduced immune surveillance against tumor cells [62,67].

### 5.3. Crioglobulinemic Disease

HCV-related cryoglobulinemic vasculitis (CV) occurs in approximately 15% of HCV-infected patients, with a broad spectrum of clinical manifestations ranging from fatigue and arthralgia to more serious complications with renal and neurological involvement [68]. In a recent international multicenter study on 913 patients with HCV-related CV, viral eradication with DAAs was associated with remission of vasculitis in 87.4% of patients 35 months after treatment. Among the patients who relapsed, 100% and 85.2% had skin and kidney manifestations, respectively. Peripheral neuropathy was present in 81.7% of relapsing patients. Independent baseline risk factors associated with cryoglobulinemic vasculitis relapse were male sex, skin ulcers, kidney involvement at baseline, and peripheral neuropathy at the end of the DAA treatment [69]. A systematic review by Danishwar et al., which included data from 19 studies on persistence and recurrence of CV after the HCV cure with DAAs, reported a complete clinical response in 63.7% to 90.2% of DAA-treated patients. Relapses were detected in 4–18% of patients. Biomarkers associated with higher incidence of persistence or recurrence of CV were identified (INFL3-rs12979860, ARNTL-rs648122, RETN-rs1423096, and SERPINE1-rs6976053), but need further validation [70]. Interestingly, in a recent multicentric prospective study on data from the Italian Platform for the Study of Viral Hepatitis Therapy cohort, the authors reported self-reported clinical deterioration after an initial response in 49.6% of patients (median time of deterioration, 19 months) and that the rate of patients without any deterioration was 58% and 41% at 12 and 24 months, respectively [71]. HCV eradication may not therefore correspond to a persistent improvement of symptoms, and clinical response can fluctuate [71].

### 5.4. Chronic Kidney Disease

There is an increased incidence of chronic kidney disease (CKD) and proteinuria in HCV-infected patients [72], as is shown in a meta-analysis of nine longitudinal studies which included 1,947,034 patients [73]. Furthermore, cryoglobulinemic vasculitis induces the development of membranoproliferative glomerulonephritis [74].

Frequent side effects and the poor efficacy of interferon-based treatments discouraged the use of IFN-based therapy in patients with end-stage kidney failure. Conversely, with DAAs, viral clearance can be attained in 92–100% of HCV patients with CKD. In November 2019, the US Food and Drug Administration allowed the use of sofosbuvir-velpatasvir in patients with renal disease, including those with an estimated glomerular filtration rate ≤ 30 mL/min and those on dialysis, providing a solution to the unmet need of treatment for HCV infection in patients with decompensated cirrhosis and concurrent advanced CKD [75]. Viral eradication leads to indirect benefits in patients with CKD by improving cardiovascular disease and reducing liver-related complications in patients with a coexisting hepatic disease [76]. There is evidence that viral clearance with DAAs may slow the progression of kidney impairment [77]. Because DAAs have only recently been permitted for the treatment of patients with advanced CKD, we still cannot estimate their impact on end-stage kidney failure. A HCV cure with DAAs has been found to reduce the incidence of cryoglobulinemic glomerulonephritis in a large retrospective cohort study of more than 45,000 HCV-treated patients [78].

### 5.5. Coagulopathy

Cirrhotic patients have a disrupted hemostatic profile, characterized by an increased risk of both bleeding and thrombosis [79]. The hypercoagulability is at least partly due to increased plasma levels of factor VIII and decreased protein C [80]. Treatment with DAAs appears to reverse these alterations in patients classified as Child–Pugh A by restoring the balance between pro- and anticoagulant factors, with more notable results appearing 12 weeks after the end of therapy [81].

### 5.6. Non-Hodgkin Lymphoma

Several studies and a metanalysis support the benefit of HCV eradication in patients with non-Hodgkin lymphoma (NHL) [82], even though there are other studies with contrasting results [78]. The metanalysis by Peveling-Oberhage et al. included 254 HCV-infected patients with NHL who underwent antiviral treatment. A strong association was found between SVR and lymphoma response (83% response rate, 95% > CI, 76–88%) compared to a failure in achieving SVR (53% response rate, 95% > CI, 39–67%, *p* = 0.0002), justifying the use of antivirals as a first-line treatment in patients who do not need immediate conventional treatment [82].

## 6. Impact on Survival and Quality of Life

A prospective, multi-center study on 1601 North American patients showed that viral clearance was associated with significant improvement in fatigue, sleep, stomach pain, and functional well-being, and these results were maintained at 12 months after treatment completion [83]. Cirrhotic patients experienced the greatest improvements in functional well-being. Patients enrolled in clinical trials with DAAs were also evaluated on long-term patient-reported outcomes, showing that improvements were durable in patients with compensated cirrhosis (up to three and a half years), but declined after two years in patients with decompensated cirrhosis [84]. These results are necessarily limited by the lack of a control arm.

## 7. Follow-Up after Eradication

According to current guidelines, the key concepts to retain on patient follow-up indications after SVR with DAAs are the following [2]:Patients with no to moderate fibrosis (meta-analysis of histological data in viral hepatitis–METAVIR–score F0#x2013;F2), no comorbidities, and those who do not have high-risk behavior can be safely discharged from surveillance, as the risk of liver-related complication is very low, and follow-up is not cost-effective.Patients with advanced fibrosis (METAVIR score F3) or cirrhosis are advised to undergo surveillance for HCC every six months.Patients with less advanced fibrosis but with coexisting risk factors (alcohol use disorder, obesity, and/or type 2 diabetes) should continue a personalized follow-up.Patients with pre-treatment esophageal varices and patients with liver stiffness > 20 kPa and platelets < 150,000/µL should undergo endoscopic assessments after SVR.It is important to monitor patients for reinfection, and retreat if needed.

In addition, other observations are presented based on current evidence (resumed in Figure 1):

HCV eradication has a positive impact on HCC surveillance with alpha-fetoprotein (AFP). In fact, inflammation can increase AFP levels leading to false-positive test results. Consequently, the performance of serum AFP is heavily influenced by ALT levels, and AFP surveillance is less useful in patients with ALT > 40 UI/L (AUC of AFP for HCC 0.91 with ALT < 40 UI/L vs. 0.76 with ALT > 40 UI/L) [85].New algorithms based on factors identifying the liver disease status (e.g., albumin, alpha-fetoprotein, liver stiffness measurement) and risk of progression (e.g., alcohol consumption) and age (a strong driver of carcinogenesis) have been developed to stratify patients according to the risk of liver cancer development [34] in order to avoid non-cost-effective surveillance. Further studies are needed to validate these results.Adherence to post-SVR follow-up decreased over the long term; therefore, patients should be encouraged to maintain their regular schedule of hospital visits if indicated [37].HCV eradication with DAAs reduces the risk of major cardiovascular events, improves insulin resistance, and is associated with high rates of remission of cryoglobulinemic vasculitis. The impact on advanced chronic kidney disease is yet to be defined.Spleen stiffness could have a role in predicting HCC development and in the assessment of CSPH after SVR.The controversy regarding the benefit of DAA-induced viral eradication after successful HCC treatment appears to be finally resolved: DAA treatment is not associated with higher HCC recurrence rates, although patients treated for HCC should undergo a close surveillance even after SVR.

## 8. Conclusions

The advent of DAAs has profoundly revolutionized the treatment of HCV-infected patients, allowing patients with more advanced stages of liver disease and comorbidities to be cured. There are many open questions related to patient management after viral eradication with DAAs, such as which could be the most reliable non-invasive tool to predict liver-related complications, or to what extent viral eradication reduces the risk of liver disease progression in the long term. There is growing evidence of the impact of DAA-based treatment on CSP, HCC development, and recurrence, as well as on extrahepatic manifestations of HCV infection which will allow us to set up more cost-effective follow-up protocols. Future research should focus on the personalization of follow-up care based on individual risk.

## Figures and Tables

**Figure 1 jcm-12-02195-f001:**
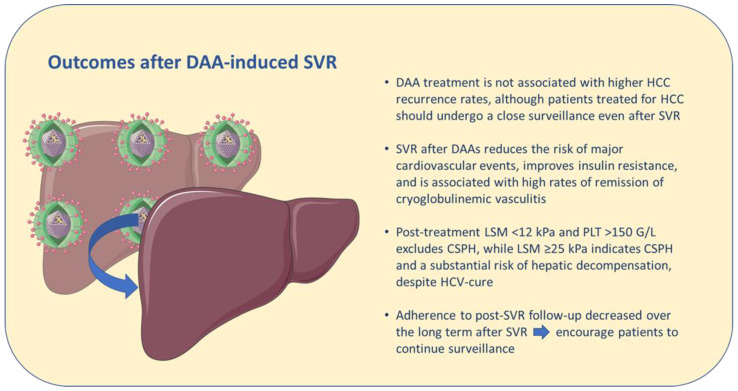
Key recent findings on outcomes after DAA-induced SVR. The figure was partly generated using Servier Medical Art, provided by Servier, licensed under a Creative Commons Attribution 3.0 unported license.

## Data Availability

Not applicable.

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
