# Peer review of "Outcomes and Follow-Up after Hepatitis C Eradication with Direct-Acting Antivirals"

_jcm, 2023, doi:10.3390/jcm12062195_

Round 1
Reviewer 1 Report
The authors review the topics on the outcomes and follow-up after hepatitis C eradication with Direct-Acting Antivirals (DAA). DAAs dramatically changed the outcomes of HCV-infected patients, but there are many unsolved questions, including the impacts on CSPH, HCC development, HCC recurrence, and extrahepatic manifestations. The authors concisely summarize the current reports on these topics. This manuscript is well written and cites appropriate papers. There are only minor points to be addressed.
Specific points.
1. Line 44. ‘ this particular patient population ‘ is unclear. What does this population mean?
Use more specific word.
2. Line 72. MELD score. Cite a reference.
3. Line 81. HVPG Don’t use abbreviation without definition.
4. Lines 170, 186, 195, and 252. the Authors → the authors
5. Line 241. in in → in
6. Line 321. >40 UI/l → >40UI/L
Author Response
We thank the editors and the reviewers for their valuable comments that helped to significantly improve our manuscript. Following their suggestions, we added requested data and revised the manuscript accordingly. Below, you will find a point-to-point reply to the editors’ and reviewers’ comments.
Reviewer 1
The authors review the topics on the outcomes and follow-up after hepatitis C eradication with Direct-Acting Antivirals (DAA). DAAs dramatically changed the outcomes of HCV-infected patients, but there are many unsolved questions, including the impacts on CSPH, HCC development, HCC recurrence, and extrahepatic manifestations. The authors concisely summarize the current reports on these topics. This manuscript is well written and cites appropriate papers. There are only minor points to be addressed.
Specific points.
- Line 44. ‘ this particular patient population ‘ is unclear. What does this population mean? We thank the reviewer for his suggestion, we have now modified the test as requested
Use more specific word.
- Line 72. MELD score. Cite a reference. We thank the reviewer for his suggestion, we have now added the reference as requested
- Line 81. HVPG Don’t use abbreviation without definition. We thank the reviewer for his suggestion, we have now modified the test as requested
- Lines 170, 186, 195, and 252. the Authors → the authors; we thank the reviewer for the suggested correction, we have now modified the test as requested
- Line 241. in in → in; we thank the reviewer for the suggested correction, we have now modified the test as requested
- Line 321. >40 UI/l → >40UI/L; we thank the reviewer for the suggested correction, we have now modified the test as requested
Reviewer 2 Report
Journal of Clinical Medicine
Outcomes and follow-up after Hepatitis C eradication with Direct-Acting Antivirals
Erica Nicola Lynch and Francesco Paolo Russo
In the literature review, the authors analyzed the impact of viral eradication with DAAs in different aspects and summarized the main indications for optimal follow-up care of HCV patients treated with DAAs. The issue is of relevant importance and the manuscript is suitable for publication in JCM. However, the authors did not classify the type of literature review. My concerns are listed below.
Line 35. For many years, interferon (IFN)-based therapy, the only available treatment for HCV infection, offered unsatisfying results, with an unfavorable tolerability profile.
Line 56 - In 2011, the first two DAAs were introduced, marking the beginning of revolutionary advances in HCV treatment. With DAAs, SVR rates reached >95% for all genotypes in non-cirrhotic patients, and 80-90% in cirrhotic 57 patients
- Please insert some information about the role of NS3/4A protease inhibitors boceprevir and telaprevir launched a new therapeutic era for HCV and the reasons why they were replaced by a new generation of DAAs.
The aim of this review is to analyze the impact of viral eradication with DAAs on clinically significant portal hypertension (CSPH),
- What kind of literature review? Narrative literature review, systematic literature review, integrative literature review reviews
Line 72 - EASL
- Write by extent. In the entire manuscript, there are several abbreviations without meaning by extent. This practice is difficult for the non-specialist reader and transforms the manuscript into a boring text.
75-77 - Please provide a short paragraph on the goal of WHO for HCV eradication.
Line 81- Successful treatment of HCV infection with interferon-based regimens reduces HVPG (written by extent) hepatic venous pressure gradient
Line 90 of liver decompensation, HCC, or LT treated (LT???)
Line 127 Liver stiffness measurement (LSM).
240-241 - eradication with DAAs was associated with remission of vasculitis in 87.4% of patients 35 months after treatment vs 12.6% of relapses, which involved the skin and kidney in (in) 100 and 85.2% of cases
- English revision is needed.
Line 254: HCV eradication may not, therefore, correspond to a persistent clinical improvement, and clinical response (s) can fluctuate.
- Improvement of what?
There are many open questions related to patient management after viral 342 eradication with DAAs, as which could be the most reliable non-invasive tool to predict 343 liver-related complications, or to what extent viral eradication reduces the risk of liver 344 disease progression in the long-term, or, finally, how to personalize follow-up care based 345 on individual risk.
- There are so many long paragraphs. The paragraph should contain up to 40 words. The entire manuscript should be English revised.
Author Response
We thank the editors and reviewers for their valuable comments that helped to significantly improve our manuscript. Following their suggestions, we added requested data and revised the manuscript accordingly. Below, you will find a point-to-point reply to the editors’ and reviewers’ comments.
Reviewer 2
In the literature review, the authors analyzed the impact of viral eradication with DAAs in different aspects and summarized the main indications for optimal follow-up care of HCV patients treated with DAAs. The issue is of relevant importance and the manuscript is suitable for publication in JCM. However, the authors did not classify the type of literature review. My concerns are listed below.
1. Line 35. For many years, interferon (IFN)-based therapy, the only available treatment for HCV infection, offered unsatisfying results, with an unfavorable tolerability profile.
2. Line 56 - In 2011, the first two DAAs were introduced, marking the beginning of revolutionary advances in HCV treatment. With DAAs, SVR rates reached >95% for all genotypes in non-cirrhotic patients, and 80-90% in cirrhotic 57 patients
Please insert some information about the role of NS3/4A protease inhibitors boceprevir and telaprevir launched a new therapeutic era for HCV and the reasons why they were replaced by a new generation of DAAs.
We agree with the reviewer and we have now added a paragraph as requested
3. The aim of this review is to analyze the impact of viral eradication with DAAs on clinically significant portal hypertension (CSPH),
- What kind of literature review? Narrative literature review, systematic literature review, integrative literature review reviews
We thank the reviewer for this comment, we have now specify the type of review
4. Line 72 - EASL
Write by extent. In the entire manuscript, there are several abbreviations without meaning by extent. This practice is difficult for the non-specialist reader and transforms the manuscript into a boring text
We agree with the reviewer and we have now added the missing meaning by extent of the abbreviations.
5. 75-77 - Please provide a short paragraph on the goal of WHO for HCV eradication.
We thank the reviewer for his suggestion, we have added a paragraph as requested
6. Line 81- Successful treatment of HCV infection with interferon-based regimens reduces HVPG (written by extent) hepatic venous pressure gradient
We thank the reviewer for his suggestion, we have added the definition as requested
7. Line 90 of liver decompensation, HCC, or LT treated (LT???)
We thank the reviewer for his question, we have added the definition as requested
8. Line 127 Liver stiffness measurement (LSM).
We thank the reviewer for his suggestion, we have modified as requested
9. 240-241 - eradication with DAAs was associated with remission of vasculitis in 87.4% of patients 35 months after treatment vs 12.6% of relapses, which involved the skin and kidney in (in) 100 and 85.2% of cases
- English revision is needed.
We thank the reviewer for his suggestion, we have modified as requested
10. Line 254: HCV eradication may not, therefore, correspond to a persistent clinical improvement, and clinical response (s) can fluctuate.
Improvement of what?
We thank the reviewer for his suggestion, we have specified as requested
11. There are many open questions related to patient management after viral eradication with DAAs, as which could be the most reliable non-invasive tool to predict liver-related complications, or to what extent viral eradication reduces the risk of liver disease progression in the long-term, or, finally, how to personalize follow-up care based on individual risk.
- There are so many long paragraphs. The paragraph should contain up to 40 words. The entire manuscript should be English revised.
We thank the reviewer for his comment, the entire manuscript has been revised by an english native speacker.
Round 2
Reviewer 2 Report
The authors filled in all the issues raised in my comments and the manuscript was well-improved and is suitable for publication in the present form.